# Comparing the Quality of Primary Care between Public and Private Providers in Urban China: A Standardized Patient Study

**DOI:** 10.3390/ijerph18105060

**Published:** 2021-05-11

**Authors:** Min Su, Zhongliang Zhou, Yafei Si, Sean Sylvia, Gang Chen, Yanfang Su, Scott Rozelle, Xiaolin Wei

**Affiliations:** 1School of Public Administration, Inner Mongolia University, Hohhot 010070, China; 111989029@imu.edu.cn; 2School of Public Policy and Administration, Xi’an Jiaotong University, Xi’an 710049, China; 3School of Risk & Actuarial Studies and CEPAR, University of New South Wales, Sydney, NSW 2052, Australia; yafei.si@unsw.edu.au; 4Gillings School of Global Public Health, University of North Carolina at Chapel Hill, Chapel Hill, NC 27599, USA; sean.sylvia@unc.edu; 5Monash Business School, Monash University, Clayton, VIC 3800, Australia; chengang1029@gmail.com; 6School of Medicine, University of Washington, Seattle, WA 98195, USA; yfsu@uw.edu; 7Freeman Spogli Institute for International Studies, Stanford University, Stanford, CA 94305, USA; rozelle@stanford.edu; 8Dalla Lana School of Public Health, University of Toronto, Toronto, ON M5T 3M7, Canada

**Keywords:** primary care, quality, public CHCs, private CHCs, standardized patient, China

## Abstract

Previous studies have been limited by not directly comparing the quality of public and private CHCs using a standardized patient method (SP). This study aims to evaluate and compare the quality of the primary care provided by public and private CHCs using a standardized patient method in urban China. We recruited 12 standardized patients from the local community presenting fixed cases (unstable angina and asthma), including 492 interactions between physicians and standardized patients across 63 CHCs in Xi’an, China. We measured the quality of primary care on seven criteria: (1) adherence to checklists, (2) correct diagnosis, (3) correct treatment, (4) number of unnecessary exams and drugs, (5) diagnosis time, (6) expense of visit, (7) patient-centered communication. Significant quality differences were observed between public CHCs and private CHCs. Private CHC physicians performed 4.73 percentage points lower of recommended questions and exams in the checklist. Compared with private CHCs, public CHC providers were more likely to give a higher proportion of correct diagnosis and correct treatment. Private CHCs provided 1.42 fewer items of unnecessary exams and provided 0.32 more items of unnecessary drugs. Private CHC physicians received a 9.31 lower score in patient-centered communication. There is significant quality inequality in different primary care models. Public CHC physicians might provide a higher quality of service. Creating a comprehensive, flexible, and integrated health care system should be considered an effective approach towards optimizing the management of CHC models.

## 1. Introduction

Primary care should be first-contact, continuous, comprehensive, coordinated, and patient-centered in order to respond to the health needs of the population [1,2]. The World Health Organization (WHO) declaration in 2008 was launched to build a primary care-oriented system, which emphasized the principles of responsiveness, quality of care, government accountability, social justice, sustainability, participation, and intersectionality [3,4]. International experiences have shown that a primary care-oriented system is able to achieve more equitable and better health outcomes because such care is less costly to individuals and more cost-effective to society [5]. Countries in different regions of the world have implemented comprehensive reforms towards primary care-oriented health system after the WHO declaration in 2008 [6,7,8,9].

In urban areas of China, community health centers (CHCs) are the main primary care facilities [10,11]. CHCs are required to have clinical physicians, public health doctors, managerial and assistant health care staff, and also on-site nurses, telephone access, and evening and weekend clinics [12,13]. Health care staff in all CHCs are paid a fixed salary, plus a floating compensation that is primarily determined by the total income of the center [13,14]. Since 2011, the Chinese government has encouraged private investment in healthcare delivery partly to promote reforms in the public sectors by introducing competition from the private sector [15]. Thus, two main models of CHCs in urban China have formed: public and private CHCs. Public CHCs are owned and managed by local government, which are components of the public health system in China and regarded as nonprofit health care facilities [13]. The most notable feature of public CHCs is their operation under the *Separation of Revenue and Expenditure* policy. The policy is that the revenue generated by public CHCs should be turned over to the local government, including fees generated from medical treatment and drug sales. However, costs incurred by the public CHCs are paid by the local government, including premises, equipment, facilities, and staff remuneration. In contrast, private CHCs are managed by a host hospital, which is usually regarded as an affiliated department within the host hospital and operates as an outreach clinic. Furthermore, private CHCs, as financially self-sufficient institutions, are allowed to make a profit through operation [10]. In order to provide more accessible and equitable primary care for all by 2020, China implemented a US$125 billion healthcare reform plan in 2009 with a critical component of facilitating CHCs to act as the first contact for patients in urban [16,17]. In 2011, the central government called for the training of general practitioners and established general-practitioner teams to provide primary medical and preventive care for the local residents. Although many policy efforts have been made to help CHCs and their development, the low quality and confidence crisis disinclined patients to seek health care in CHCs.

Recent decades have witnessed a wide range of researches investigating the quality of primary care. Some scholars have evaluated the primary care quality based on patient perception and recall-based surveys. For example, by using the Primary Care Assessment Tool developed by Johns Hopkins Primary Care Center [18,19], Wei et al. assessed the patient-perceived quality of CHCs in three Chinese megacities from five key attributes (first-contact, continuous, comprehensive, coordinated, and patient-centered) [12,13]. However, this method is subjective, subject to recall bias, and primarily focuses on patient perception rather than actual clinical ability. Recently, standardized patients presenting fixed disease cases have been regarded as the gold standard to measure healthcare quality [20,21,22,23,24,25]. For example, Das et al. selected unstable angina, asthma and dysentery to assess the quality of primary care in India [23]. They found low levels of primary care and substantial-quality gaps between rural and urban clinics. Sylvia et al. used unstable angina and dysentery to evaluate the quality of primary care in rural China [21]. The study reported poor quality care in village clinics and township health centers. However, they do did not directly compare the quality of public and private CHCs using the standardized patient method. A previous study described and compared primary care under public and private CHCs using a face-to-face patient’s experience survey from four aspects: services, organization, financing, and human resources. They found that the ownership and management of a community health center significantly influence the service it provides, and private CHCs may be in a disadvantageous position to provide services. However, the study focused on patients’ subjective experiences and offered little information on physicians’ actual capacity and quality during the clinical practice. Importantly, the subjective bias and recall bias might be a crucial problem in understanding primary care quality under different models.

This study aims to evaluate and compare the quality of primary care provided by public and private CHCs using the standardized patient method (SP). Compared with private CHCs, public CHCs tend to have a favorable development environment due to the more considerable public funding [11]. Thus, we hypothesize that the quality of primary care provided by public CHCs is higher than that of private CHCs. In anticipation, our findings would inform policymakers in China and akin countries’ tentative provision of public/private healthcare.

## 2. Materials and Methods

### 2.1. Study Design and Setting

This study was conducted in urban Xi’an, China, from 17 to 28 August 2017, and 30 July to 10 August 2018. A standardized patient is a healthy individual recruited from the local community and trained to portray an actual patient’s historical, physical and emotional features in a standardized way to collect the communication information between physician and patient [21,23]. We recruited SPs from the local community, who were provided with a financial allowance and, to ensure the retention of SPs, a bonus for those who continued participation until the end of the study. Additionally, SPs were reimbursed for the visit fee. Overall, 12 SPs were selected after the announcement of recruitment, including eight female SPs and two male SPs in 2017 and seven female SPs and one male SP in 2018. The 12 SPs participated in rigorous training before visiting physicians at CHCs. The training focused on preparing SPs to represent their assigned disease cases to providers in a consistent and unsuspicious manner [21].

Xi’an is the capital of Shaanxi province, located in the northwest of China, with a population of 8.8 million in an area of over 9983 km^2^; 73.4% of residents in Xi’an lived in urban areas in 2018. Per capital gross regional product (GRP) was US$ 12,103 in 2018. We selected all urban districts of Xi’an as our samples. There were 65 CHCs in the selected areas, and all of them were invited to participate, and two CHCs consented. Two of them were declined because they only provided public health, not basic medical health. Finally, this study consisted of 63 CHCs (53 public CHCs vs. 10 private CHCs).

### 2.2. Standardized Patient Procedure

#### 2.2.1. Case Selection and Scenario

SPs were trained to portray unstable angina and asthma. Unstable angina and asthma were chosen because (1) these diseases had a high incidence in China (approximately 8% and 2%, respectively, for people aged 50 and older), (2) these diseases are easier to be portrayed without obvious symptoms and with low risk of invasive examinations, and (3) these diseases were selected from SP cases employed in previous studies internationally and within China [23,24]. For unstable angina, a 50-year-old patient reported having chest pain recently and it felt like there was something heavy pressing on the chest. For asthma, a 40-year-old patient reported a problem with breathing and stated that last night it became terrible [23,24].

#### 2.2.2. Script and Checklist

Physicians and professors with rich experience in conducting SP studies were invited to develop the scripts. The scripts did cover all possible questions and examinations a physician may ask or perform during the interaction. Each script included (1) a detailed background story for each case, (2) an opening statement of symptoms portrayed by SPs, (3) an illness history presented in a question-and-answer format, and (4) possible examinations and treatments. The checklists of specific cases provide recommended questions and examinations that a physician should ask or perform during the interactions (The full checklists are given in the Appendix A).

#### 2.2.3. Recruitment and Training of Standardized Patient

SPs were recruited from the local communities to make sure that they were representative of actual patients commonly diagnosed by the primary care providers. SPs were chosen following these essential criteria [19,20]: (1) SPs must be in good physical conditions without confounding symptoms; (2) non-physicians were recruited as SPs because physician-SP knowledge and behaviors may affect diagnosis and treatment of the counterpart physician; (3) a reasonable level of intelligence, and memory and communication ability are essential; and (4) SPs should match the predesigned cases in age.

SPs participated in three-day training by a team consisting of professors, medical experts, and investigators. The training included: (1) details of case scenarios and scripts were explained to SPs, (2) recordings of interactions between SPs and physicians obtained from our pilot study were presented to SPs, (3) role-playing and one-on-one training were used to help SPs to understand and memorize the scripts and portray the cases, (4) some principles and excuses were taught to SPs to respond to the physician’s questions not designed in scripts or to avoid examinations; and (5) the SPs’ performances were assessed.

#### 2.2.4. Standardized Patient Visit

Four SPs were randomly assigned and independently visited each CHC (two SPs portraying unstable angina and the other two SPs portraying asthma). SPs could not choose physicians by themselves and they must be seen by whoever would have seen them had they been a common patient once entering the practical settings [21]. SPs wore a concealed recording device to record the interactions between physicians and SPs.

#### 2.2.5. Investigator Training and Standardized Patient Exit Survey

Twelve postgraduate students attended three workshops to train as investigators to complete the exit survey of SPs immediately after the visit based on the visit recordings. Three methods were used to collect data on the SP–physician interactions [21]. Firstly, SPs wore a concealed recording device, which allowed us to accurately score interactions without recall bias. Secondly, SPs were administered a case-specific ‘debriefing survey’ upon exiting CHCs. This survey covered the interaction as well as the SP’s impressions of the providers. Finally, SPs purchased all medications prescribed and paid all fees to collect information on medications dispensed and fees charged.

#### 2.2.6. Evaluating the Quality of the Primary Care

We measured the quality of the primary care on seven criteria: (1) adherence to clinical checklists, (2) accuracy of diagnosis, (3) appropriateness of treatment, (4) the number of unnecessary exams and drugs, (5) diagnosis time, (6) expense of visit, (7) patient-centered communication (PCC), including a total score of PCC, and the sub-score of PCC1, PCC2, and PCC3.

(1) Adherence to the checklist: the Sylvia et al. Chinese adaptation of the Das et al. clinical checklists for unstable angina and asthma were used to determine adherence to the clinical checklists. Adherence was defined as the proportion of recommended questions that were asked and recommended examinations that were performed during the SP consultation. Adherence was treated as a continuous scale. Full checklists could be found in Appendix A.

(2) Accuracy of diagnosis: SPs were instructed to ask physicians directly at the end of the session if a diagnosis was initially offered. The diagnosis was classified as ‘correct’ if physicians gave any one of the correct diagnose according to the pre-determined criteria. For unstable angina, the correct diagnose included: unstable angina, angina, or coronary heart disease; the correct diagnose of asthma included: asthma, or allergic asthma.

(3) Appropriateness of treatment: The treatment was defined as correct if the provider prescribed any one of the correct medications. SPs were referred to tertiary hospitals or secondary hospitals was also an appropriate treatment for unstable angina following WHO guidelines [24,25]. Unstable angina is an intermediate state between acute myocardial infarction and stable angina. Due to its unique pathophysiological mechanism and clinical characteristics, if it is not properly treated in a timely fashion, the patient may develop acute myocardial infarction. In order to avoid delays in patient visit due to the low quality of the primary care facilities (e.g., lack of inspection equipment and qualified physicians), referral was also an appropriate treatment for unstable angina.

(4) Unnecessary exams and drugs: An unnecessary exam and/or drug was defined as the exam and/or drug that was prescribed by the physicians that was unnecessary or even harmful. We included all necessary exams and drugs in our predesigned checklist and thus the item was regarded as unnecessary if it did not fall into the checklist [21,22]. SPs received all physical examinations and rejected all laboratory examinations according to the scripts. Therefore, unnecessary exams and drugs were counted when one test was provided or planned to be provided by physicians in the study. We used the number of unnecessary exams and drugs in this study.

(5) Diagnosis time: Diagnosis time was defined as the length of time that physicians spent on consulting with SPs and regarded as a proxy for provider effort [21,22]. We used consultation time measured by minutes in the study.

(6) Expense of visit: Medical expense mainly included exam and drug expenses in the primary care setting. The expense of each visit was calculated as the sum of the visit fee, all prescribed drugs, and all exams (performed and planned). SPs paid the visit fee and purchased all prescribed drugs.

(7) Patient-centered communication: patient-centered care was measured by the nine-item patient perception of patient-centeredness (PPPC) rating scale [26,27,28,29,30]. Using a five-point Likert scale answered by SPs, the patient perception of patient-centeredness was evaluated on three dimensions: exploring the disease and illness experiences (PCC1), understanding the whole person (PCC2), and finding common ground (PCC3) [26,27,28,29,30]. The detailed definitions of each quality dimension are given in the Appendix A (Appendix A).

### 2.3. Statistical Analysis

Our analysis unit was the interaction between physician and SP. Our econometric specification was:(1)$$y_{ijt}=\beta_{1}Model_{ijt}+\beta_{2}X_{ijt}+\beta_{3}W_{ijt}+\delta_{j}+\varphi_{t}+\mu +\varepsilon_{ijt}$$
where $y_{ijt}$ represented the quality of the primary care indicator that was analyzed in CHC *i* district *j* on day *t*. Model represented CHC models, including public CHCs and private CHCs. $X_{ijt}$ was a set of the observable demographic correlates of the physicians. $W_{ijt}$ was a set of the observable character of CHCs. $\delta_{j}$ indicated the fixed effect of survey regions (districts) and μ indicated disease fixed effects. $\varphi_{t}$ indicated the fixed effect of survey time (day–month–year). $\varepsilon_{ijt}$ was the error term. Robust standard errors were clustered at the community health center level. Ordinary least-squares regression models were used for the continuous variables and logistic regression models were used for the categorical variables. All analyses were carried out using Stata (vision 15).

### 2.4. Sensitivity Analysis

We examined the consistency of the results regarding the following: (1) using two different cases, and (2) controlling for different potential confounding factors (i.e., with and without missing data in the regression models).

### 2.5. Ethics

The ethics approval was obtained by the Ethics Committee of Xi’an Jiaotong University Health Science Center (approval number: 2015-406). We were approved to record the interactions between physicians and SPs using a concealed recording device. Written consents were obtained from physician and the director of each CHC along with a face-to-face survey approximately three months prior to SP visits.

## 3. Results

### 3.1. Basic Characteristics

This study included 492 interactions across two cases. Among them, interactions from public CHCs accounted for 414 out of 492 (84.15%), and interactions from CHCs within health alliance accounted for 436 out of 492 (88.62%). Most physicians were female, aged between 40 and 50 years old. Most SPs were female. The average working experience of physicians was 22.87 years. Most were practicing (assistant) physicians (95.82%) and had an educational level at bachelor’s degree or below (59.83%). Interactions differed by health alliance status, and physician working experience between public CHCs and private CHCs; however, there were no differences in terms of other variables. More details could be found in Table 1.

### 3.2. The Quality of Primary Care Provided by Public CHCs and Private CHCs

Table 2 described the quality of the primary care provided by public and private CHCs. On average, 31.89% (SD 16.06) and 34.55% (SD 15.73) of the clinical checklists were completed for public CHCs and private CHCs, respectively. The proportion of interactions resulting in the correct diagnosis was 43.24% and 48.72%, and the correct pharmaceutical prescription was 25.60% and 16.67% for public and private CHCs. Unnecessary examinations were less commonly prescribed by public CHC than private CHC physicians (mean 0.86 items, SD 1.00 vs. mean 1.17 items, SD 1.26). In contrast, unnecessary drugs were more commonly prescribed by public CHC than private CHC physicians (mean 0.47 items, SD 0.84 vs. mean 0.31 items, SD 0.69). The average consultation length was 6.12 min (SD 4.27) for public CHC physicians and 6.65 min (SD 5.67) for private CHC physicians. The total medical costs were lower in public CHCs than private CHCs (mean 34.31 CNY, SD 40.39 vs. mean 38.67 CNY, SD 45.76, respectively). The average PCC score was 23.15 (SD 6.18) for interactions in public CHCs and 23.55 (SD 6.55) for private CHCs.

### 3.3. Association between the CHC Models and the Quality of Primary Care

Table 3 shows the association between the CHC models and the quality of primary care. After adjustments were made for CHC characteristics, sociodemographic characteristics of providers, SP gender, and potential fixed effects, significant-quality differences were observed between public CHCs and private CHCs. Specially, private CHC physicians performed 4.73 lower percentage points of recommended questions and exams than public CHC physicians (*p* < 0.1). Private CHC physicians presented 28.70 lower percentage points of correct diagnosis (*p* < 0.01), and 70.00 lower percentage points of correct treatment than public CHC physicians (*p* < 0.01). Private CHC physicians provided 1.42 fewer items of unnecessary exams (*p* < 0.01). Private CHCs provided 0.32 more items of unnecessary drugs than public CHCs (*p* < 0.1). SPs in private CHCs paid 0.82 CNY more for average visit. Total score in patient-centered communication was 9.31 lower in private CHCs (*p* < 0.01). PCC1 was 3.37 lower in private CHCs (*p* < 0.01). PCC2 was 0.48 lower in private CHCs (*p* < 0.01). PCC3 was 5.46 lower in private CHCs (*p* < 0.01). The result also shows that the CHCs without health alliance had significantly 3.69 percentage points higher adherence to the clinical checklist (*p* < 0.01), provided 0.30 more items of unnecessary exams (*p* < 0.05), and 0.86 more items of unnecessary drugs (*p* < 0.01), spent 1.75 fewer minutes on consultation (*p* < 0.01), and paid 26.18 CNY more for an average visit (*p* < 0.01). SPs in CHCs without health alliance received 1.81 higher scores in patient-centered communication (*p* < 0.01), received 2.30 higher scores in PCC1 (*p* < 0.01), received 0.18 lower scores in PCC2 (*p* < 0.01), and received 0.31 lower score in PCC3.

### 3.4. Sensitivity Analysis Outcomes

Table 4 shows the sensitivity analysis results for asthma. The results were close to our original analysis results. For example, private CHC physicians performed 1.33 lower percentage points of recommended questions and exams than public CHC physicians (*p* > 0.05). Private CHC physicians presented 56.80 lower percentage points of correct diagnosis (*p* < 0.01), and 51.40 lower percentage points of correct treatment than public CHC physicians (*p* < 0.01). Private CHC physicians provided 2.17 fewer items of unnecessary exams (*p* < 0.01). The total score in patient-centered communication was 8.38 lower in private CHCs (*p* < 0.01). PCC1 was 0.58 lower in private CHCs (*p* > 0.05). PCC2 was 0.84 lower in private CHCs (*p* < 0.01). PCC3 was 6.97 lower in private CHCs (*p* < 0.01).

Table 5 shows the sensitivity analysis results for unstable angina. Private CHC physicians performed 13.70 lower percentage points of recommended questions and exams than public CHC physicians (*p* > 0.05). Private CHC physicians presented 43.20 lower percentage points of correct diagnosis (*p* > 0.05), and 61.20 lower percentage points of correct treatment than public CHC physicians (*p* < 0.05). Private CHCs provided 1.32 more items of unnecessary drugs than public CHCs (*p* < 0.1). SPs in private CHCs paid 47.56 CNY more for an average visit. The total score in patient-centered communication was 12.32 higher in private CHCs (*p* < 0.10). PCC1 was 2.31 higher in private CHCs (*p* > 0.05). PCC2 was 0.05 higher in private CHCs (*p* > 0.05). PCC3 was 9.96 lower in private CHCs (*p* < 0.01). More details can be found in Table 5.

Table 6 shows the sensitivity analysis results by controlling different potential confounding factors. The results show that private CHC physicians presented 70.00 lower percentage points of correct diagnosis (*p* < 0.01), and 81.90 lower percentage points of correct treatment than public CHC physicians (*p* < 0.01). Private CHC physicians provided 1.20 fewer items of unnecessary exams (*p* < 0.01). The total score in patient-centered communication was 5.03 lower in private CHCs (*p* < 0.10). PCC1 was 1.39 lower in private CHCs (*p* > 0.05). PCC2 was 0.26 lower in private CHCs (*p* > 0.05). PCC3 was 3.39 lower in private CHCs (*p* < 0.01). More details can be found in Table 6.

## 4. Discussion

This study is the first to evaluate and compare the quality of primary care between public and private providers using an SP approach in urban China. The SP approach, which focused on physician’s actual behavior, capacity, and quality during the clinical practice recording by a recording device, rather than patient’s experience and satisfaction with health care delivery, minimizes subjective bias and recall bias due to the sociodemographic variations and patient expectation. Previous studies have used the SP approach to measure health care quality under different health care systems and different types of providers [10,11,12,19]. This study added to the evidence suggesting that the quality of the primary care measured by the SP approach was related to the management model of the CHCs in western urban China with less advantages in social-economic development. In 2007, China’s Ministry of Health identified 36 nationally contacted cities in the surveillance data of community health service system construction. In this national program, there were 308 public CHCs (82.13%) and 67 private CHCs (17.87%). It could be seen that the breakdown of public CHCs and private CHCs in our study reflected the ratio of the two models in the whole area and urban China more broadly.

Compared with private CHCs, public CHC providers were more likely to adhere to the clinical checklist, giving a higher proportion of correct diagnosis and correct prescriptions. Furthermore, public CHC providers earned a higher score on patient-centered communication. Based on our in-depth qualitative review and previous study, these differences likely reflect the inherent financial incentives and human resources in the two management models. Possible reasons for differences in the quality of care could be explained using the ecological model (e.g., macro-, meso-, micro-level). At the macro level, China’s new healthcare reform in 2009 has encouraged private investment to promote reforms in the public sectors by creating competition [15,31]. However, the health system in China was still dominated by the public sectors [31]. In such a context, the key challenges of private sectors included low patient recognition, lack of qualified physicians, and weaker support from local authorities in issues such as land use, taxation, and government subsidies. At the meso level, the ownership may be a possible explanation related to differences in the quality of care. While the local government directly supports public CHCs, private CHCs receive limited funding from the local government through the host hospital. More importantly, all the government funding is allocated to the host hospital before being distributed to CHCs [10,11,12]. The host hospital might deduct from the CHC funding to compensate hospital management or other costs [8,9]. Moreover, the local government is responsible for evaluating the performance of public CHCs on an annual basis. Based on the assessment, the best public CHCs would get an extra 10% more government funding. At the micro level, human resources are another reason to explain the quality differences between different CHC models. For example, staff in public CHCs are provided with a secured position (namely *Bianzhi* quotas) and benefits, while *Bianzhi* quotas in private CHCs are far fewer so that private CHCs need to hire more physicians without quotas [12]. A physician without quotas means lower income, fewer opportunities for promotion, and insecurity. Previous evidence has shown that staff in public CHCs have better educational levels and more opportunities for training [12]. Our in-depth qualitative review found that private CHCs act as outreach clinics of host hospitals. Private CHCs are smaller in the size of the organization and human recourses. They also need to undertake public health tasks, which distract private CHCs from providing medical care. However, more empirical studies are required in order to test the specific reasons.

The performance of the providers that we observed in urban Xi’an were substantially better than what was found in a previous SP study in rural areas of the same province and used the same unstable angina case script [21]. In terms of the diagnostic process, the proportion of recommended questions plus exams was 32.31% in urban community health centers in the current study compared to just 17.86% in village clinics and 27.38% in township health centers in rural areas of the province. The rate of correct diagnosis was 44.11% in urban community health centers compared with 25.00% in village clinics and 58.00% in township health centers. These may be due to differences in human resources in primary care facilities. Previous surveys have shown significant differences in human resources in rural and urban health facilities. For example, one study showed that, as of 2014, the proportion of practicing doctors and assistant practicing doctors who had a bachelor’s degree or higher in community health centers was 37%, compared with only 11.9% in township health centers and 1.9% in village clinics [30].

Our study also found that health alliance was another important factor associated with better primary care quality because of the close cooperation and collaboration among the tertiary hospitals, secondary hospitals, and CHCs within the health alliance. Health alliance is helpful to sink the high-quality health resources of tertiary hospitals and secondary hospitals into primary health facilities. For example, the tertiary hospitals undertake the tasks of management guidance, technical assistance, and personnel training for primary care facilities. In order to give regular training, a package of training activities for physicians from primary and secondary hospitals is provided through offering on-site training, giving advice, providing clinical services alongside local practitioners, and training general medical practitioners in tertiary hospitals’ training centers [32,33,34,35]. Additionally, primary care physicians within the health alliance could have free study opportunities in tertiary hospitals, which is advantageous for primary care physicians to study the standard and advanced diagnosis and treatment technology. Another possible reason for the positive effect of health alliance is the guidelines and clinical pathways. Health alliance actively promotes the discipline guidelines and clinical pathways, and all health facilities abided by the unified diagnosis and treatment guideline and operation standard. Thus, they gradually realize the unified, standardized, and homogeneous diagnosis and treatment of disease, which is also conducive to standardizing the health behavior of primary care physicians.

## 5. Conclusions

This study is the first to evaluate and compare the quality of primary care between public and private providers using the SP approach in urban China. This study added to the evidence suggesting that the quality of the primary care measured by the SP approach was related to the management model of the CHCs in urban China. In conclusion, interactions between physicians and SPs indicate that the quality of primary care in urban China is poor overall. Compared with private CHCs, public CHC providers were more likely to adhere to the clinical checklist, giving a higher proportion of correct diagnosis and correct treatment. Public CHC providers earned a higher score on patient-centered communication. Creating a comprehensive, flexible, and integrated health care system should be considered an effective approach towards optimizing the management models of CHCs. The specific reasons for this difference need to be further explored in the future. This study should be interpreted in the context of several potential limitations. Firstly, the SP approach is limited to cases that are easier to portray without obvious physical symptoms and low-risk invasive examinations. Secondly, due to the study area, the conclusion may have limited generalizability. Thirdly, the conclusion that differences in quality of care between public CHCs and private CHCs likely reflect financial incentive aspects was based on our in-depth qualitative review and previous study. We did not have data on the financial incentive aspects of public CHCs and private CHCs. Fourthly, since SPs were healthy, they obviously presented no symptoms, so this may be underestimating the correct diagnosis rate and correct treatment rate. Finally, our study only concentrated on correlation analysis. Bearing the limitations in mind, this study has highlighted several strategies that might be helpful to improve the quality of primary care. First, the clinical capacity of the primary care providers should be strengthened by providing sufficient education and training. Second, it is important to increase primary care provider’s salary. Thirdly, the management model of private CHCs should be optimized by improving financial transparency and human resource development between private CHCs and the holding hospitals. Finally, health alliance should be considered towards improving primary care quality.

## Figures and Tables

**Table 1 ijerph-18-05060-t001:** Characteristics of interactions between physicians and SPs.

Variables	Total		Public CHCs		Private CHCs		*p* Value
	No.	%	No.	%	No.	%	
**Health alliance**							
Yes	436	88.62	358	86.47	0	0.00	0.001
No	56	11.38	56	13.53	78	100.00	
N	492		414		78		
**SP gender**							
Female	411	83.54	343	82.85	68	87.18	0.344
Male	81	16.46	71	17.15	10	12.82	
N	492		414		78		
**Physician age group**							
Age < 30	28	5.69	26	6.28	2	2.56	0.283
30 ≤ Age < 40	122	24.80	107	25.85	15	19.23	
40 ≤ Age < 50	181	36.79	148	35.75	33	42.31	
Age ≥ 50	161	32.72	133	32.13	28	35.90	
N	492		414		78		
**Physician gender**							
Female	268	54.47	222	53.62	46	58.97	0.384
Male	224	45.53	192	46.38	32	41.03	
N	492		414		78		
**Physician working experience** (years), mean, S.D.	22.87	12.09	22.22	11.98	27.93	11.98	0.021
N	239		212		27		
**Physician education**							
Bachelor’s degree and above	96	40.17	88	41.51	8	29.63	0.236
Bachelor’s degree or below	143	59.83	124	58.49	19	70.37	
N	239				27		
**Practicing (assistant) physician**							
Yes	229	95.82	202	95.28	27	100.00	0.249
No	10	4.18	10	4.72	0	0.00	
N	239		212		27		

Note: (1) N refers to the number of interactions between physicians and SPs; for variables such as health alliance, SP gender, physician age, there is no missing data; thus, there are 492 interactions in total. For variables such as physician working experience, physician education, and practicing (assistant) physician, they have missing data; thus, there are 239 interactions except the missing data. (2) χ2 tests were used for categorical variables; t tests were used for continuous variables. (3) In our study, physicians represent general practitioners in CHCs. Practicing (assistant) physician represents a kind of professional qualification of general practitioners. (4) Source: the author’s calculation.

**Table 2 ijerph-18-05060-t002:** Quality of the primary care provided by public and private CHCs.

Variables	Definition	Total	Public CHCs	Private CHCs	*p* Value
**(1) Adherence to checklist** (average proportion, SD)	Proportion of recommended questions plus exams	32.31 (16.02)	31.89 (16.06)	34.55 (15.73)	0.844
**(2) Correct diagnosis** (N, %)	Physicians give correct diagnose after consultation	217 (44.11)	179 (43.24)	38 (48.72)	0.371
**(3) Correct treatment** (N, %)	Physicians prescribed at least one correct drug. Referring SPs to tertiary hospitals or secondary hospitals was also an appropriate treatment for unstable angina following WHO guidelines	119 (24.19)	106 (25.60)	13 (16.67)	0.091
**(4) Unnecessary exam and drug (mean, SD)**					
Unnecessary exams	Number of unnecessary or harmful examinations	0.91 (1.05)	0.86 (1.00)	1.17 (1.26)	0.035
Unnecessary drugs	Number of unnecessary or harmful drugs	0.45 (0.82)	0.47 (0.84)	0.31 (0.69)	0.063
**(5) Diagnose time (mean, SD)**	Consultation time (minutes)	6.21 (4.52)	6.12 (4.27)	6.65 (5.67)	0.440
**(6) Total cost (mean, SD)**	Expenditure for the visit (CNY)	35.00 (41.26)	34.31 (40.39)	38.67 (45.76)	0.133
**(7) Patient Centered Communication (mean, SD)**					
PCC	Total score of PCC (score, 0–49)	23.22 (6.24)	23.15 (6.18)	23.55 (6.55)	0.482
PCC1	Exploring both the disease and illness experience (score, 0–29)	12.24 (4.04)	12.09 (4.05)	13.00 (3.95)	0.801
PCC2	Understanding the whole person (score, 0–3)	0.79 (0.64)	0.78 (0.64)	0.82 (0.62)	0.753
PCC3	Finding common ground (score, 0–17)	10.19 (3.60)	10.28 (3.62)	9.73 (3.48)	0.699
N		492	414	78	

Note: (1) χ2 tests were used for categorical variables; *t* tests were used for continuous variables. (2) Source: the author’s calculation.

**Table 3 ijerph-18-05060-t003:** The effect of CHC models on the quality of health care.

Variables	(1)	(2)	(3)	(4)	(5)	(6)	(7)	(8)	(9)	(10)	(11)
	Adherence to Checklist	Correct Diagnosis	Correct Treatment	Number of Unnecessary Exams	Number of Unnecessary Drugs	Diagnosis Time	Total Cost	PCC	PCC1	PCC2	PCC3
Private CHCs	−4.734 *	−0.287 ***	−0.700 ***	−1.416 ***	0.321 *	−0.306	0.820	−9.312 ***	−3.368 ***	−0.479 ***	−5.464 ***
	(2.527)	(0.094)	(0.099)	(0.182)	(0.165)	(0.583)	(5.097)	(1.393)	(0.721)	(0.150)	(0.779)
Non-Health alliance	3.685 ***	−0.027	−0.077	0.295 **	0.858 ***	−1.745 ***	26.181 ***	1.814 ***	2.304 ***	−0.183 ***	−0.307
	(1.155)	(0.068)	(0.054)	(0.137)	(0.139)	(0.643)	(3.032)	(0.591)	(0.331)	(0.064)	(0.369)
Male SP	0.450	0.125 **	0.063	0.410 ***	−0.253 **	−0.180	3.862	0.512	−0.413	0.0004	0.925 **
	(1.383)	(0.060)	(0.0487)	(0.153)	(0.119)	(0.481)	(4.521)	(0.651)	(0.359)	(0.080)	(0.389)
Male Physician	−1.342	−0.010	−0.004	−0.043	0.065	−0.419	−3.519	−0.077	−0.231	−0.037	0.191
	(1.354)	(0.066)	(0.044)	(0.131)	(0.091)	(0.553)	(4.865)	(0.754)	(0.446)	(0.081)	(0.433)
30 ≤ Age < 40	1.626	−0.030	−0.119	0.264	0.138	−0.432	20.260 *	0.993	0.613	0.207	0.173
	(3.724)	(0.134)	(0.103)	(0.264)	(0.196)	(1.345)	(10.345)	(1.871)	(1.076)	(0.189)	(1.040)
40 ≤ Age < 50	−0.215	−0.060	−0.122	0.159	0.107	−0.823	17.293 *	−0.061	0.141	0.183	−0.386
	(3.622)	(0.134)	(0.098)	(0.262)	(0.181)	(1.241)	(10.172)	(1.797)	(1.068)	(0.192)	(1.025)
Age ≥ 50	−0.731	−0.132	−0.068	0.072	0.218	0.185	22.165 *	0.311	0.011	0.218	0.082
	(4.010)	(0.136)	(0.102)	(0.274)	(0.182)	(1.299)	(12.074)	(1.967)	(1.139)	(0.203)	(1.111)
Case	20.306 ***	−0.365 ***	−0.250 ***	0.553 ***	0.086	0.974 **	6.642	1.324 **	2.796 ***	0.085	−1.557 ***
	(1.087)	(0.045)	(0.049)	(0.098)	(0.079)	(0.452)	(4.271)	(0.561)	(0.366)	(0.060)	(0.313)
N	492	492	492	492	492	492	492	492	492	492	492
Coefficient of determination	0.577	0.336	0.277	0.348	0.291	0.271	0.368	0.302	0.400	0.271	0.264

Note: (1) Eleven models were used to evaluate the effect of CHC models on the quality of health care. (2) Ordinary least-squares regression models were used for the continuous variables and logistic regression models were used for the categorical variables. (3) The fixed effects included: survey regions (districts); disease cases (unstable angina and asthma); survey time (day–month–year). (4) Standard errors in parentheses. (5) * *p* < 0.1, ** *p* < 0.05, *** *p* < 0.01. (6) Source: the author’s calculation.

**Table 4 ijerph-18-05060-t004:** The effect of CHC models on the quality of health care for Asthma.

Variables	(1)	(2)	(3)	(4)	(5)	(6)	(7)	(8)	(9)	(10)	(11)
	Adherence to Checklist	Correct Diagnosis	Correct Treatment	Number of Unnecessary Exams	Number of Unnecessary Drugs	Diagnosis Time	Total Cost	PCC	PCC1	PCC2	PCC3
Private CHCs	−1.334	−0.568 ***	−0.514 ***	−2.166 ***	−0.262	−1.003	−5.841	−8.382 ***	−0.581	−0.836 ***	−6.965 ***
	(3.591)	(0.177)	(0.166)	(0.138)	(0.198)	(1.393)	(9.929)	(2.219)	(1.159)	(0.243)	(1.414)
Non-Health alliance	1.222	−0.047	−0.283 ***	1.291 ***	0.338 ***	−1.050	26.090 ***	−1.512	1.217	0.275 **	−3.003 ***
	(2.643)	(0.097)	(0.082)	(0.189)	(0.087)	(0.908)	(7.458)	(1.352)	(0.916)	(0.128)	(0.669)
Male SP	0.067	−0.016	0.086	−0.008	−0.179 *	−0.362	0.220	5.364 ***	0.876	0.040	4.447 ***
	(3.188)	(0.122)	(0.121)	(0.265)	(0.102)	(0.887)	(7.534)	(1.766)	(1.017)	(0.161)	(0.971)
Male Physician	−0.251	0.087	0.049	0.208	0.171	−0.629	1.578	0.831	−0.149	−0.027	1.006
	(2.228)	(0.101)	(0.087)	(0.233)	(0.106)	(0.869)	(6.665)	(1.056)	(0.724)	(0.122)	(0.680)
30 ≤ Age < 40	−2.525	0.082	−0.189	0.117	0.085	−0.758	15.270	−1.487	−0.932	0.154	−0.709
	(5.090)	(0.217)	(0.153)	(0.340)	(0.246)	(2.188)	(15.710)	(2.562)	(1.700)	(0.331)	(1.341)
40 ≤ Age < 50	−5.240	−0.042	−0.158	−0.168	−0.060	−2.600	6.342	−2.710	−1.679	−0.063	−0.968
	(4.756)	(0.211)	(0.147)	(0.331)	(0.264)	(2.101)	(16.340)	(2.473)	(1.615)	(0.310)	(1.406)
Age ≥ 50	−3.837	−0.167	−0.196	−0.267	−0.030	0.161	11.580	−2.358	−1.089	−0.055	−1.214
	(4.552)	(0.219)	(0.153)	(0.405)	(0.247)	(1.886)	(19.490)	(2.592)	(1.528)	(0.344)	(1.631)
N	245	245	245	245	245	245	245	245	245	245	245
Coefficient of determination	0.429	0.347	0.487	0.392	0.573	0.446	0.495	0.476	0.454	0.390	0.499

Note: (1) Eleven models were used to evaluate the effect of CHC models on the quality of health care. (2) Ordinary least-squares regression models were used for the continuous variables and logistic regression models were used for the categorical variables. (3) The fixed effects included: survey regions (districts); disease cases (unstable angina and asthma); survey time (day–month–year). (4) Standard errors in parentheses. (5) * *p* < 0.1, ** *p* < 0.05, *** *p* < 0.01. (6) Source: the author’s calculation.

**Table 5 ijerph-18-05060-t005:** The effect of CHC models on the quality of health care for unstable angina.

Variables	(1)	(2)	(3)	(4)	(5)	(6)	(7)	(8)	(9)	(10)	(11)
	Adherence to Checklist	Correct Diagnosis	Correct Treatment	Number of Unnecessary Exams	Number of Unnecessary Drugs	Diagnosis Time	Total Cost	PCC	PCC1	PCC2	PCC3
Private CHCs	−13.700	−0.432	−0.612 **	0.332	1.316 ***	3.097	47.560 **	12.320 *	2.312	0.047	9.959 ***
	(17.610)	(0.671)	(0.280)	(1.087)	(0.424)	(1.876)	(18.870)	(6.278)	(4.627)	(0.965)	(1.991)
Non-Health alliance	4.065	−0.006	0.107	−0.612 **	1.510 ***	−2.858 ***	35.710 ***	5.442 ***	3.307 ***	−0.508 ***	2.643 ***
	(2.648)	(0.078)	(0.084)	(0.250)	(0.320)	(0.984)	(8.651)	(1.258)	(0.862)	(0.157)	(0.660)
Male SP	4.545 **	0.158 *	0.085	0.640 ***	−0.277	−0.249	3.693	−0.356	−0.403	0.010	0.038
	(2.197)	(0.086)	(0.055)	(0.201)	(0.181)	(0.687)	(6.365)	(0.986)	(0.576)	(0.106)	(0.635)
Male Physician	−2.200	−0.101	−0.055	−0.275	−0.129	0.093	−15.520	−1.614	−0.658	−0.104	−0.852
	(2.771)	(0.087)	(0.061)	(0.197)	(0.179)	(1.003)	(9.373)	(1.426)	(0.847)	(0.157)	(0.799)
30 ≤ Age < 40	6.690	−0.210	−0.110	0.728	0.164	−0.031	24.320	1.857	2.010	0.191	−0.345
	(6.313)	(0.194)	(0.119)	(0.495)	(0.275)	(2.003)	(15.750)	(2.772)	(1.665)	(0.263)	(1.659)
40 ≤ Age < 50	5.811	−0.237	−0.121	0.781 *	0.329	0.377	20.440	1.252	1.732	0.398	−0.878
	(6.034)	(0.191)	(0.111)	(0.440)	(0.266)	(1.953)	(15.470)	(2.794)	(1.606)	(0.248)	(1.581)
Age ≥ 50	3.550	−0.208	0.055	0.613	0.482 *	0.200	30.510 **	2.252	1.207	0.491 *	0.554
	(6.625)	(0.194)	(0.115)	(0.460)	(0.287)	(1.795)	(14.700)	(2.916)	(1.776)	(0.251)	(1.537)
N	0.247	247	247	247	247	247	247	247	247	247	247
Coefficient of determination	0.571	0.448	0.454	0.560	0.411	0.408	0.591	0.525	0.509	0.456	0.449

Note: (1) Eleven models were used to evaluate the effect of CHC models on the quality of health care. (2) Ordinary least-squares regression models were used for the continuous variables and logistic regression models were used for the categorical variables. (3) The fixed effects included: survey regions (districts); disease cases (unstable angina and asthma); survey time (day–month–year). (4) Standard errors in parentheses. (5) * *p* < 0.1, ** *p* < 0.05, *** *p* < 0.01. (6) Source: the author’s calculation.

**Table 6 ijerph-18-05060-t006:** The effect of CHC models on the quality of health care by controlling different potential confounding factors.

Variables	(1)	(2)	(3)	(4)	(5)	(6)	(7)	(8)	(9)	(10)	(11)
	Adherence to Checklist	Correct Diagnosis	Correct Treatment	Number of Unnecessary Exams	Number of Unnecessary Drugs	Diagnosis Time	Total Cost	PCC	PCC1	PCC2	PCC3
Private CHCs	0.868	−0.700 ***	−0.819 ***	−1.195 ***	−0.073	1.895	2.032	−5.034 *	−1.387	−0.262	−3.385 **
	(4.768)	(0.175)	(0.066)	(0.316)	(0.289)	(1.397)	(8.546)	(2.968)	(1.661)	(0.276)	(1.390)
Non-Health alliance	10.590 ***	−0.059	0.029	0.907 **	−0.236	−5.163 ***	−5.071	3.156 **	3.587 ***	−0.115	−0.316
	(2.746)	(0.117)	(0.087)	(0.344)	(0.177)	(1.380)	(8.812)	(1.567)	(0.741)	(0.150)	(1.058)
Male SP	2.285	0.075	−0.016	0.629 ***	−0.458 ***	−0.263	3.241	0.820	0.293	0.032	0.496
	(2.379)	(0.112)	(0.094)	(0.181)	(0.162)	(0.763)	(7.603)	(1.214)	(0.672)	(0.152)	(0.734)
Male Physician	−1.620	−0.043	0.068	−0.294	0.045	−0.496	−2.452	0.166	−0.222	−0.093	0.481
	(2.692)	(0.111)	(0.087)	(0.314)	(0.184)	(1.257)	(9.274)	(1.632)	(0.825)	(0.202)	(0.858)
30 ≤ Age < 40	4.017	−0.401 *	−0.076	−0.269	0.090	4.544 **	14.940	0.263	1.042	0.293	−1.072
	(8.878)	(0.221)	(0.221)	(0.587)	(0.479)	(2.241)	(18.070)	(4.479)	(2.052)	(0.493)	(2.491)
40 ≤ Age < 50	0.291	−0.438 **	−0.049	−0.613	0.059	4.266 *	14.030	−1.049	−0.326	0.401	−1.124
	(9.079)	(0.204)	(0.225)	(0.643)	(0.470)	(2.484)	(19.010)	(4.469)	(0.297)	(0.501)	(2.543)
Age ≥ 50	0.887	−0.484 **	−0.032	−0.818	0.086	5.702 **	12.870	0.705	0.550	0.360	−0.206
	(9.970)	(0.215)	(0.238)	(0.710)	(0.468)	(2.519)	(20.620)	(5.102)	(2.333)	(0.528)	(2.864)
Physician working experience	−0.111	−0.008	0.002	−0.001	−0.003	−0.045	−0.555	−0.108	−0.032	−0.011	−0.066
	(0.136)	(0.007)	(0.004)	(0.021)	(0.010)	(0.066)	(0.540)	(0.104)	(0.044)	(0.009)	(0.070)
High school and above	−6.572	−0.305	0.057	−0.357	−0.144	1.102	−25.290 **	−1.880	−1.434	−0.089	−0.356
	(4.043)	(0.202)	(0.135)	(0.424)	(0.299)	(1.608)	(12.070)	(3.254)	(1.344)	(0.204)	(2.079)
Practicing (assistant) physician	−1.060	0.259	0.096	−0.431	0.125	−2.915	−1.612	−0.247	−0.660	0.033	0.380
	(6.591)	(0.214)	(0.174)	(0.556)	(0.311)	(3.335)	(11.980)	(3.280)	(1.807)	(0.291)	(1.790)
Case	23.020 ***	−0.378 ***	−0.135 **	0.417 ***	0.186	0.243	7.371	1.960 *	3.279 *	−0.034	−1.284 **
	(1.590)	(0.080)	(0.060)	(0.144)	(0.113)	(0.731)	(7.692)	(1.003)	(0.555)	(0.108)	(0.623)
N	239	239	239	239	239	239	239	239	239	239	239
Coefficient of determination	0.683	0.503	0.428	0.537	0.425	0.349	0.448	0.374	0.504	0.319	0.362

Note: (1) Eleven models were used to evaluate the effect of CHC models on the quality of health care. (2) Ordinary least-squares regression models were used for the continuous variables and logistic regression models were used for the categorical variables. (3) The fixed effects included: survey regions (districts); disease cases (unstable angina and asthma); survey time (day–month–year). (4) Standard errors in parentheses. (5) * *p* < 0.1, ** *p* < 0.05, *** *p* < 0.01. (6) Source: the author’s calculation.

## Data Availability

The datasets generated for the current study are available from 245 the corresponding author on reasonable request.

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
