# Peer review of "Comparing the Quality of Primary Care between Public and Private Providers in Urban China: A Standardized Patient Study"

_ijerph, 2021, doi:10.3390/ijerph18105060_

Round 1
Reviewer 1 Report
Thank you for giving me the opportunity to read this manuscript.
1) The authors write: “We hypothesize that the quality of primary care provided by public CHCs is higher than that in private CHCs”. Why? Please explain in one or two sentences.
2) The study was conducted in 2017 and 2018 (we are already in 2021…). Why wasn’t the study done in one and the same moment? There is one year of difference between research samples. Many things can change during one year… Are these two samples are comparable?
3) The authors write: “The 12 SPs participated in rigorous training before visiting physicians at CHCs…”. What does it mean? Please add one or two sentences about the training when you mention about the training, even if it is explained in details in section 2.2.
4) The authors write: “The previous study indicated private CHCs had greater clinical capacity because they are directly managed by public hospitals”. Please give the appropriate references of these previous study. And explain what does it mean: “private CHCs they are directly managed by public hospitals”.
5) I believe that the results are very interesting, but the authors must do an effort to make attractiveness this extraction of information. They are difficult to assimilate.
6) The conclusion is underdeveloped and very short. It does not really tie other manuscript elements together in any meaningful way. Theoretical and Practical Contributions should be clearly evidenced.
I suggest as well to move the part concerning limitations to the Conclusion section. Please add there also the future research which are not specified.
7) The paper require proper editing in terms of English language.
I wish the authors all the best in developing their manuscript.
Reviewer 2 Report
This study evaluated and compared the quality of the primary care provided by public and private CHCs using standardized patient method. They study was able to demonstrate that the quality of primary care provided by public CHCs was higher than that provided in private CHC in urban Xi'an, China.
The finding of this study will inform policy makers in China.
The study methodology was described well. However there is a need to further define the difference between a physician and a practicing (assistant) physician as the difference will not be clear to readers who do not work in the health system in China. What are the qualification differences and job descriptions and roles? Table1 presents physician education as middle school and below and Highschool and above. Is this the normal educational levels for Physicians in China? This needs to be explained
The results and analysis are clearly presented. The discussion was based on the findings and the deductions were sound.
The authors recognise their limitations. The recommendations will be useful for policy changes
Reviewer 3 Report
This paper evaluates the quality of primary care between public and private providers using standardized patient approach in urban China. I find that the topic is important and the research method is appropriate. To improve the paper, I think that the authors need to elaborate why the quality of primary care differs depending on the management model of the CHCs in urban China.
In the discussion section, authors explained the reason related to differences in the quality of primary care using the previous qualitative review. Readers can understand how and why financial incentive aspects influence physicians’ performance if authors would explain the findings of qualitative study in detail. Possible explanations related to differences in the quality of care could be reorganized and revised using ecological model (e.g., macro-, meso-, micro-level). In that way, international readers can better understand how the primary care works in China.
Lastly, some typos are found and it is necessary to proofread the draft thoroughly. For example, “blow” should be changed into “below” in Table 1.
